# Clinical outcomes associated with schistosome infection and alcohol use: A systematic scoping review

**Bethany Lyne[1], Max M. Lang[1], Sarah Lewington[1,2], Goylette F. Chami [1]***

**1** Nuffield Department of Population Health, University of Oxford, Oxford, United Kingdom, **2** Health Data Research UK Oxford, University of Oxford, Oxford, United Kingdom

* goylette.chami@ndph.ox.ac.uk

## Abstract

### Background

Schistosomiasis and alcohol use are major, co-occurring risk factors for liver disease in low and middle-income countries. However, their interactions and shared disease outcomes remain poorly understood.

### Methods

We conducted a scoping review to understand how schistosome infection and alcohol use influence author-defined health outcomes. A systematic search was carried out on 3 March 2025 using the following databases: PubMed (1946 – present), Embase (1974 – present), Web of Science (1964 – present), Global Index Medicus (1901 – present), and Global Health (1973 – present). Quality of studies was assessed descriptively by assessing bias and confounding.

### Results

The search yielded 2358 articles, with 21 studies eligible for synthesis. Most studies (76.2%, 16/21) focused on current *S. mansoni* infection, and the same proportion treated alcohol use as a binary variable. The most frequently reported clinical outcomes were periportal fibrosis (42.9%, 9/21), and biomarkers (19%, 4/21), including serum iron markers. For hepatic outcomes, both synergistic and antagonistic pathways are plausible, while for iron-related outcomes, an antagonistic relationship was supported. Cross-sectional studies were most common (66.7%, 14/21), limiting the ability to make assessments on the temporal relationships between exposures and clinical outcomes. Selection bias was the most frequently reported source of bias (28.6%, 6/21), and only four (19%) studies reported both adjusted and unadjusted analyses, allowing for an assessment of confounding.

**Data availability statement:** All extracted data has been provided as supplementary material.

**Funding:** Funding from Nuffield Department of Population Health as a DPhil Studentship to Beth Lyne; and from the UKRI Grant [EP/X021793/1] to GFC. The funders had no role in study design, data collection and analysis, decision to publish, or preparation of the manuscript.

**Competing interests:** The authors have declared that no competing interests exist.

## Conclusions

Co-occurring schistosomiasis and alcohol use has been associated with more severe liver pathology, presenting a significant public health concern in endemic areas. Current literature focuses on schistosome-related pathology, where alcohol use acts as a modifying factor. Future research should prioritise longitudinal designs with standardised definitions of alcohol use to better understand interactions between these two exposures for liver outcomes.

## Author summary

Schistosomiasis is a chronic parasitic disease that is prevalent predominantly in sub-Saharan Africa, but also found in South-East Asia and South America. In these regions, schistosomiasis and heavy alcohol use are two major and frequently co-occurring causes of liver disease. It is currently unclear whether these two exposures cause disease through additive pathways, or whether they interact biologically to accelerate disease progression. This review aims to identify disease outcomes associated with concurrent schistosome infection and alcohol use, explore the potential biological pathways, and assess the quality of the existing evidence on schistosome infection and alcohol use as concurrent exposures. For liver injury, the literature suggests both antagonistic and synergistic pathways are plausible, but evidence from human studies is inconsistent. However, there was a lack of studies which can address the temporality of these exposures, and the current literature does not consider the changes in sequence, intensity and duration of schistosome infection and alcohol use. Future studies need to prioritise longitudinal designs which monitor people from childhood, when schistosome infection first occurs, through to adulthood when drinking behaviours develop.

## Introduction

Liver disease accounts for approximately 4% of all global deaths, with cirrhosis and hepatocellular carcinoma representing the most common and severe manifestations [1]. Schistosomiasis and alcohol use are both major causes of liver disease, and their co-distribution in low- and middle-income countries (LMICs) presents an important public health concern, despite being driven by different risk factors [2]. Schistosomiasis is a neglected tropical disease with varied disease presentations and co-occurring pathologies that affects an estimated 250 million individuals worldwide; it is caused by trematodes and transmitted during contact with contaminated freshwater [3]. Although schistosomiasis-related mortality is difficult to quantify due to limited data, recent World Health Organisation (WHO) estimates suggest approximately 11,792 deaths per year [4]. In contrast, alcohol use is influenced by a range of cultural, psychological, and socioeconomic risk factors [5,6]. According to the WHO, around 400

million people globally live with an alcohol use disorder and 4.7% of all deaths in 2019 could be attributed to an alcohol-related liver disease [7]. Fishing communities represent settings where schistosomiasis risk and alcohol use are both prevalent [8], as populations experience prolonged water contact, increasing exposure to schistosome infection, and high levels of alcohol use have been reported due to unique social and occupational contexts [9]. While the individual disease presentations associated with schistosomiasis and alcohol use have been studied separately [10–12], their potential interactions and shared disease presentations in co-exposed populations remains unexplored.

The potential for synergistic liver pathologies between schistosomiasis and alcohol use is biologically plausible. *Schistosoma mansoni*, *S. japonicum*, and *S. mekongi* cause hepatic schistosomiasis, which can manifest as periportal fibrosis (PPF) and portal hypertension [13]. Eggs laid by adult female schistosomes lodge in the portal vein of the liver, releasing antigens that trigger granuloma formation. Chronic inflammation, brought about by chronic infections, leads to PPF and disrupts the architecture of the liver, causing portal hypertension. Alcohol, after absorption into the bloodstream, also enters the liver via the portal vein. Then, alcohol dehydrogenase enzymes break the alcohol down into acetaldehyde, a highly toxic and carcinogenic compound [14]. The production of acetaldehyde, along with reactive oxygen species (ROS), play a central role in the pathogenesis of alcoholic liver disease (ALD). ALD follows a well-recognised pattern of liver damage – starting with alcoholic fatty liver (steatosis), and progressing to more severe conditions such as alcoholic steatohepatitis, cirrhosis, and eventually hepatocellular carcinoma [15]. While these pathways differ mechanistically, both result in inflammation and hepatic fibrosis, meaning there is the potential for additive or synergistic harm in co-exposed individuals that remains to be identified.

Immunological studies in mice have demonstrated that co-occurring schistosome infection and alcohol consumption lead to more severe pathology than either exposure individually [16,17]. In a study by Brandão-Bezerra et al., mice exposed to both *S. mansoni* infection and chronic alcohol intake developed greater liver damage (larger granulomas and increased fibrosis), compared to mice with only one of the exposures [16]. The authors suggest that alcohol-induced immune dysfunction may disrupt the formation of granulomas around schistosome eggs, altering disease progression. While excessive or unregulated granuloma formation contributes to pathology, granulomas also play an important protective role by trapping schistosome eggs and limiting the spread of antigens [18]. Similarly, da Rosa et al. reported that co-exposed mice exhibited greater structural and functional damage to the spleen [17], reflecting the relationship between hepatic and splenic pathology in advanced schistosomiasis. Here, they noted the decrease in both pro- and anti-inflammatory cytokines, showing that alcohol use weakens the immune response to infection. Despite these findings in murine models, the disease pathways and mechanisms in concurrent schistosome infection and alcohol use in humans remain poorly understood.

We conducted a systematic scoping review to map the existing literature on co-occurring schistosome infection and alcohol use, with a focus on their potential interactions and associated disease outcomes. The aim of this study was to assess the existing evidence on the combined effect of schistosomiasis and alcohol use on clinical outcomes in human populations. We focus on answering questions related to how co-occurring schistosome infection and alcohol use impact clinical outcomes, the biological and temporal pathways underlying disease progression, the methodological approaches used to study these interactions, and the gaps that exist in the current evidence base. The key overarching hypothesis leading into this review is that the simultaneous presence of alcohol use with schistosome infection, whether past or current, leads to worse liver outcomes and that the effect is primarily synergistic.

## Methods

### Database search and selection criteria

The full study protocol was prospectively published on 26 February 2025 on the Open Science Framework website [19]. To ensure appropriate and transparent reporting, the PRISMA extension for Scoping Reviews (PRISMA-ScR) guidelines were followed (S1 File) [20]. A systematic search was performed on 3 March 2025 across five databases: PubMed (1946

– present), Embase (via Ovid) (1974 – present), Web of Science Core Collection (1964 – present), Global Index Medicus (1901 – present), and Global Health (via Ovid) (1973 – present). The search terms used (in the PubMed/MEDLINE format) were: (Schistosom* OR Bilharzia* OR snail* fever) AND (alcohol* OR drink*). This search string was adapted for each database, and full search strategies for all databases are provided in S2 File. Results were exported to Covidence [21], where duplicates were removed, and any missed duplicates from Covidence were removed manually.

### Selection criteria

Two reviewers (BL and ML) independently screened all titles and abstracts for eligibility. Where results were eligible for full text screening, a full text of the study was obtained and all English full texts were screened by one reviewer (BL). A random 10% of the English full texts were screened by the second reviewer (ML). Disagreements at either of the screening stages were resolved through discussion between the two reviewers, and if unresolved then through a third reviewer (GFC). If a study was not in English but eligible for full text screening, a translation was sought. Reference lists of all the eligible full texts were searched for potentially eligible articles.

Studies conducting original research in any year up to the search date were eligible for inclusion. There were no restrictions on study participants or the species of schistosome. Included studies must have described both exposure to schistosome infection and alcohol use within the same individuals. Any description of alcohol use was included, including the frequency and type of alcohol consumed. Although this review was motivated by potential outcomes relating to liver disease, there was no restriction on the type or severity of clinical outcome studied to comprehensively capture all the reported associations for these exposures when they co-occur. Self-reported clinical outcomes and medical histories were included. To summarise effect sizes, studies were required to report statistical comparisons between alcohol use, schistosome infection, and clinical outcomes, or provide sufficient data to calculate these comparisons, for example through a 2 × 2 table. Studies were excluded if the case definition of the clinical outcome included schistosome infection or alcohol use as part of the diagnostic criteria, rather than treating them as independent exposures. Animal studies were not eligible, neither were ecological studies, case reports, reviews (systematic, narrative, meta-analyses), opinion articles or commentaries, or grey literature. Other study designs, including cross-sectional studies, cohorts, before-and-after studies, randomised controlled trials, and case-control studies were eligible. If a study was reported in multiple publications, the report with the most participants was considered if the two publications used the same study population, time period and geographical location. If the population, location, and number of participants was the same, the most recent publication was included.

### Data extraction and qualitative synthesis

A data extraction form and data dictionary were adapted from Ockenden and colleagues [22], and piloted using Microsoft Excel to capture study characteristics and variables specific to answering the questions of the review. Three papers [23–25], which had been used to validate the search string, were used to pilot the extraction table, and any required changes were made before extracting from all the included studies. One reviewer (BL) extracted from all the included studies, and a second reviewer (ML) verified a random 10% of the extractions. The completed data extraction table is provided in S3 File. The extracted variables were related to study characteristics and study participants, exposures, outcomes, and statistical methods used. The study characteristics were author name(s), publication year, study year, country, study setting, locality, study inclusion and exclusion criteria, study aim, study design, time points, sample size, maximum and minimum age of participants, and sex of participants. Other extracted variables included schistosome species, schistosome diagnostic, method used to gather data on alcohol use, infection prevalence, alcohol use prevalence, co-occurrence prevalence, clinical outcome as defined by the study authors, statistical method used, measure of association reported, and, if possible, what relationship the authors suggested between co-occurring schistosome infection and alcohol use in relation to the clinical outcome. At the extraction stage, types of bias and confounding that were relevant to the study were considered, whether directly reported by the study authors or inferred in this analysis. Selection bias

was identified in studies where the sample was not considered representative, for example from the use of non-random sampling methods or the recruitment of participants from settings like hospitals, where the sample would over-represent people who are already ill. Information bias was evaluated by assessing misclassification of the clinical outcome, such as when the outcome was not clearly defined or could be used inconsistently across participants. For each clinical outcome, outcome-specific confounders were identified both from the study reports and through assessment during data extraction; for example, hepatitis co-infection for liver fibrosis, or human immunodeficiency virus (HIV) viral load for renal impairment. For all studies, we also assessed whether common confounders were accounted for such as age and sex. The lack of consideration of confounding was particularly noted when studies reported unadjusted analyses.

The included studies were grouped according to their primary aims and fell into six broad categories. Risk factor studies examined associations between various exposures, including alcohol use and schistosome infection, and disease markers or clinical outcomes such as PPF and impaired renal function. Prevalence and disease burden studies measured the occurrence of schistosome infection and alcohol use, as well as the related morbidities in various populations. Some targeted specific groups, such as people living with HIV or tuberculosis, or residents of endemic communities. Diagnostic studies were classified based on their focus of assessing diagnostic accuracy or validation of diagnostic methods and tools related to disease outcomes. Treatment studies evaluated interventions, for example evaluating the impact of chemotherapy on morbidity. Clinical consequence studies focused on the manifestations and complications of schistosomiasis or alcohol-related morbidity, as well as how schistosome infection or alcohol use influences the presentation of comorbid conditions. Finally, biomarker studies investigated various biological markers, such as those related to iron status, by exploring the influence of dietary and infectious factors on disease outcomes.

In accordance with guidelines from the Joanna Briggs Institute guidance for scoping reviews [26], data were synthesised using a basic qualitative content analysis. Key themes synthesised included the types of clinical outcomes studied (and their appropriateness, as assessed by biological plausibility), the statistical methods used for analysis, and the overall quality (assessed through bias and confounding) of the study. A key focus of the synthesis was on the identification of clinical relationships and potential disease pathways described in the literature, with the aim of understanding how outcomes may be influenced through potential mechanistic pathways involving alcohol use and schistosome infection and the temporality between exposures and exposures and outcomes.

## Results

### Screening process

A flowchart of the scoping review process is presented in Fig 1. The initial database search yielded 2358 hits, of which 541 were duplicates. The number of hits retrieved from each database is provided in the S2 File. Following screening, 21 studies were extracted and included in this review. The most common reasons for exclusion when screening full texts were the absence of data on schistosome infection or alcohol consumption, lack of clinical outcome (mostly where studies sought to report the prevalence of schistosome infection), and the use of an ineligible diagnostic method for schistosome infection. The latter category comprised of studies relying on self-reported infection status as well as studies which used proxy indicators of infection, for example classifying participants as infected based on the presence of fibrosis. The full list of excluded full texts, and their reasons for exclusion, is provided in S4 File.

### Study characteristics

A summary of the key characteristics of the studies included in the review are presented in Table 1. The most frequently studied schistosome species was *S. mansoni* (80.0%, 16/20), followed by *S. japonicum* (15.0%, 3/20). Only one study focused on *S. haematobium*, and one study failed to report the species of schistosome being studied. The most commonly studied alcohol exposure was a binary variable which compared any alcohol consumption to none (42.9%, 9/21), followed

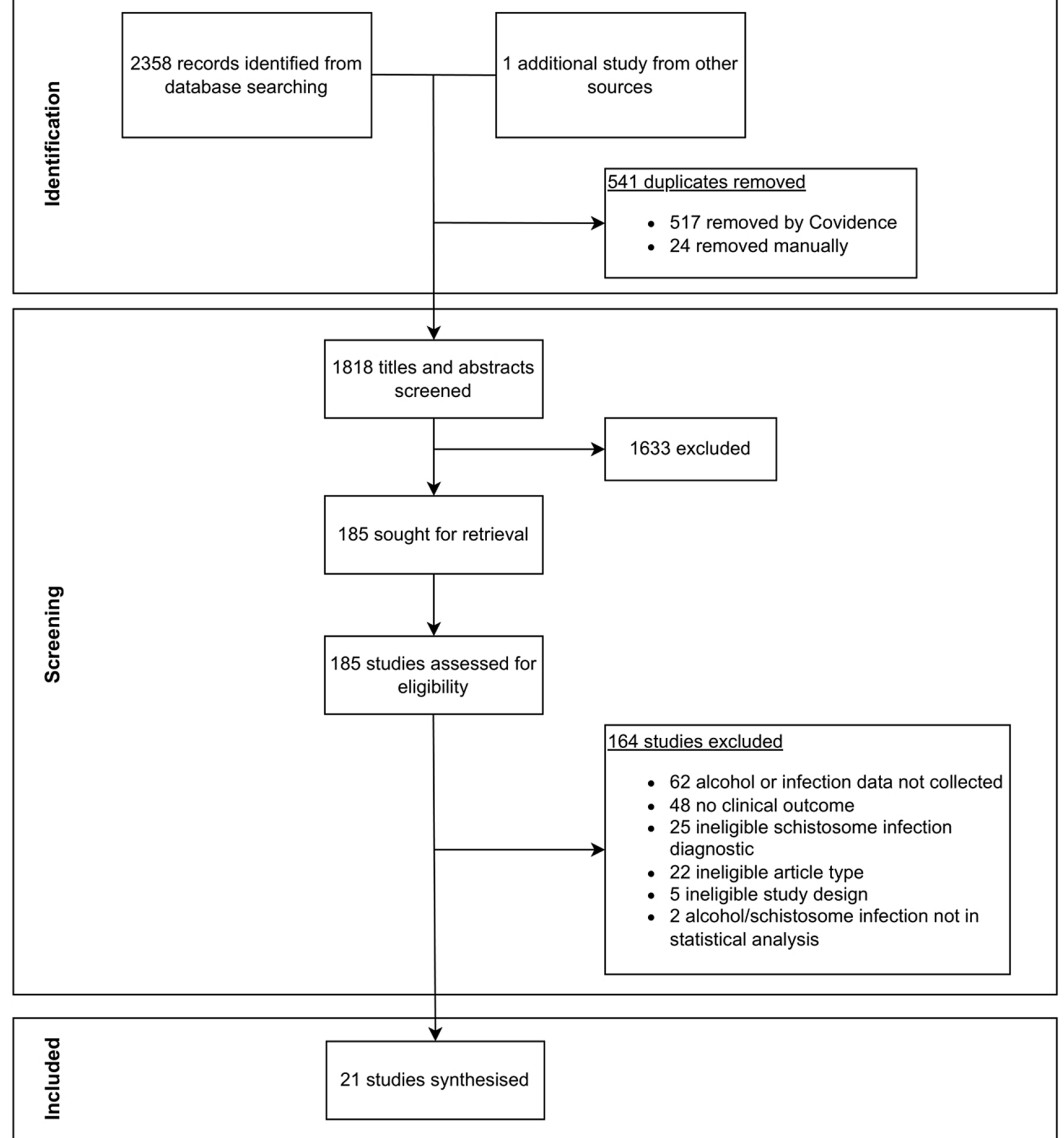

**Fig 1. PRISMA flowchart showing the study selection process.**

by seven studies (33.3%) that compared participants with high alcohol consumption to those with lower or no alcohol intake.

Nine countries across four WHO regions were represented. The most common study countries were Brazil (33.3%, 7/21), Tanzania (19.0%, 4/21), and Uganda (14.3%, 3/21). Most studies (95.2%, 20/21), made no restrictions on participant sex, and only one study was female-only. Only 13 studies reported an age range of participants, with reported ages spanning from 5 to 90 years across all the studies. Nine studies reported a mean age, and six reported a median age. The total sample size across all the included studies was 11,794, with a range of 54 – 2834. The median number of participants was 257 (IQR 100 – 655). Cross-sectional studies were the most common (66.7%, 14/21), although cohort (23.8%, 5/21), case-control (4.8%, 1/21), and RCT (4.8%, 1/21) designs also were represented.

PLOS Neglected Tropical Diseases

**Table 1. Key characteristics of the 21 studies included in the synthesis. Percentages represent the proportion of studies that reported each characteristic.**

| Characteristic | Total studies that reported | Category/Statistic | No. of studies within each category | Percentage |
|---|---|---|---|---|
| Study aims | 21 | Prevalence and disease burden | 6 | 28.6 |
| | | Risk factors | 5 | 23.8 |
| | | Treatment | 4 | 19.0 |
| | | Biomarkers | 3 | 14.3 |
| | | Clinical consequences | 2 | 9.5 |
| | | Diagnostic | 1 | 4.8 |
| Clinical outcome | 21 | Periportal fibrosis | 9 | 42.9 |
| | | Biomarkers | 4 | 19.0 |
| | | Portal hypertension | 2 | 9.5 |
| | | Susceptibility to other infections | 2 | 9.5 |
| | | Cirrhosis | 1 | 4.8 |
| | | Bladder pathology | 1 | 4.8 |
| | | Renal function | 1 | 4.8 |
| | | Cancer | 1 | 4.8 |
| Study country | 21 | Brazil | 7 | 33.3 |
| | | Tanzania | 4 | 19.0 |
| | | Uganda | 3 | 14.3 |
| | | China | 2 | 9.5 |
| | | Nigeria | 1 | 4.8 |
| | | Mali | 1 | 4.8 |
| | | Zimbabwe | 1 | 4.8 |
| | | Philippines | 1 | 4.8 |
| | | Saudi Arabia | 1 | 4.8 |
| Study design | 21 | Cross-sectional | 14 | 66.7 |
| | | Cohort | 5 | 23.8 |
| | | Case-control | 1 | 4.8 |
| | | RCT | 1 | 4.8 |
| Study setting | 21 | Community | 9 | 42.9 |
| | | Health centre | 6 | 28.6 |
| | | Hospital | 4 | 19.0 |
| | | University | 2 | 9.5 |
| Participant sex | 21 | Male and female | 20 | 95.2 |
| | | Female only | 1 | 4.8 |
| Schistosome species | 20 | *Schistosoma mansoni* | 16 | 80.0 |
| | | *Schistosoma japonicum* | 3 | 15.0 |
| | | *Schistosoma haematobium* | 1 | 5.0 |
| Schistosome diagnostic | 21 | Microscopy | 18 | 85.7 |
| | | Serology | 2 | 9.5 |
| | | Biopsy | 1 | 4.8 |
| Alcohol use assessment | 17 | Self-administered questionnaire | 11 | 64.7 |
| | | Interview | 3 | 17.6 |
| | | Self-report | 3 | 17.6 |

The most commonly reported aims of the included studies were to quantify disease prevalence and burden (28.6%, 6/21), investigate risk factors associated with the clinical outcome of interest (23.8%, 5/21), and evaluate treatment methods (19.0%, 4/21). Less frequent aims were those related to the study of biomarkers (14.3%, 3/21), clinical consequences (9.5%, 2/21), and diagnostics (4.8%, 1/21). The most frequently reported clinical outcome across the studies was PPF, addressed in 42.9% (9/21) of the studies. Biomarkers were the second most reported outcome (19.0%, 4/21), encompassing a study investigating serum endotoxin levels, as well as studies concerned with markers of iron deficiency. Portal hypertension was the outcome of interest in only two studies. Another two studies examined susceptibility to coinfections, namely HIV and tuberculosis. Additional outcomes, each represented in a single study were cirrhosis, bladder pathology, impaired renal function, and cancer.

### Varied definitions of alcohol use, schistosome infection, and diseases

Across all the studies, 76.2% (16/21) defined alcohol use as a binary variable, although they used varied reference categories. Of these, nine (56.3%, 9/16) studies compared any alcohol consumption to none (drinkers versus non-drinkers). The other seven studies (43.7%, 7/16) compared high or author-defined problem alcohol use to lower or no use, such as alcoholics versus non-alcoholics, or alcohol abuse versus no alcohol abuse. Four studies (19.0%) treated alcohol use as a categorical variable; three studies classified participants as never, ever, or regular drinkers, while the fourth study included an additional category for ex-drinkers. Only 17 (81.0%) of the studies reported information on their alcohol data collection method. Of these, 64.7% (11/17) used questionnaires completed by the participant themselves. Three studies (17.6%) conducted interviews with the study participant directly or with the household head, and in three studies (17.6%), the method was simply referred to as "self-report" without clarification as to how the data were collected. Four studies (19.0%) provided no information as to how alcohol use data were collected. Only one study clearly reported using a standardised tool, which was the WHO STEPwise Approach to Surveillance (WHO STEPS). Definitions of alcoholism, problem drinking, or regular drinking were inconsistent across studies, with most failing to clearly describe their criteria. For the five studies that did report definitions, thresholds showed large variation. One study defined alcoholism as consumption of more than 20g or 30g of alcohol per day for men and women, respectively, and another defined it as consumption of more than 60g per day for all participants. Other definitions were based on frequency, such as consuming alcohol three or more times per week. The timeframe for alcohol consumption – whether it referred to lifetime use, recent use, or another specified period – was often unclear or not reported. One study did not clearly define how they collected and coded alcohol consumption data; however, it reported a correlation coefficient with "lifetime alcohol consumption".

For schistosome infection, most studies (85.6%, 18/21) assessed infection status using microscopy. Of these, 17 (94.4%) analysed stool samples using Kato-Katz techniques and one analysed urine samples using urine filtration. All of the Kato-Katz microscopy studies, as well as one study which used tissue biopsy, defined infection as the presence of any eggs in the sample. Two studies used antigen-based diagnostics – one applied the circulating cathodic antigen (CCA) urine dipstick test, and one used a lateral flow assay to detect circulating anodic antigen (CAA). No studies assessed infection using antibody-based tests. Nearly all studies (95.2%, 20/21) treated infection as a binary variable (infected versus uninfected). Although some studies reported measures of infection intensity, only one study incorporated a categorical classification into its analysis, defining infection intensity based on *S. haematobium* egg counts from urine filtration as: no infection (0 eggs/ 10 mL), light infection (≤50 eggs/ 10 mL), heavy infection (>50 eggs/ 10 mL). Only 10 (47.6%) of the studies using microscopy reported the number of samples used to diagnose schistosome infection. Of these, six reported using one sample, two used two samples, and another two used three samples. Where multiple samples were collected, one sample was taken per day over consecutive days.

A full summary of the definitions used in each study is presented in Table 2. Diagnostic methods and case definitions used to assess clinical outcomes varied across studies. For the outcome of PPF, all studies used abdominal ultrasonography as the diagnostic method. Only one study explicitly described using point-of-care ultrasound, although the other

**Table 2. Summary of the definitions of schistosome infection, alcohol use, and outcome used in each study. Also included are the study population and age range (if reported), study exclusion criteria, number of participants, the statistical methods used in the analysis, and the association observed.**

| Study and design | Alcohol use definition | Schistosome infection definition | Outcome | Population (age range) | Exclusion criteria | Number of participants | Statistical method | Result |
|---|---|---|---|---|---|---|---|---|
| Kavishe (2021) [25]; cross-sectional | Unhealthy drinking = >2 standard drinks/day for women, >3 standard drinks/day for men. Moderate drinker = anyone else who reported currently drinking. Also had never drinker and ex-drinker. | Presence of eggs in stool sample. | Impaired renal function (eGFR <60 mL/min/1.73m²). | Adults (18+). | Under 18, pregnant, terminally ill, ART-naïve HIV-infected adults had to be willing to start ART after enrolment. | 1947 | Logistic regression, linear regression. | OR (moderate drinkers vs non-drinkers) = 0.7 (0.2–3.1). OR (unhealthy drinkers vs non-drinkers) = 1.1 (0.7–1.6). OR (schistosome infection vs no infection) = 0.8 (0.4–1.7). Only unhealthy drinking was a significant predictor of eGFR in the linear regression (coefficient: –4.7; p=0.03). |
| Malenganisho (2008) [24]; cross-sectional | Drinkers, non-drinkers (not defined). | Infection was defined as presence of eggs in stool sample. Infection intensity was defined as number of eggs per gram of stool. | PPF (liver image pattern C–F). | Participants were aged 14 and above (14 – 87 years). | None reported. | 1447 | Pearson's chi-squared, ANOVA. | There was an association between alcohol intake and liver image pattern in Msozi (p=0.03), but not in Sangabuye (p=0.06). An association was observed between infection intensity and higher PPF grades in Msozi (p=0.02), but not in Sangabuye. |
| Ocama (2017) [23]; cross-sectional | Drinkers, non-drinkers (yes/no to alcohol consumption). | Positive CCA urine test. | PPF (liver image pattern C–F). | Adults (18+) with HIV. (18 – 69 years) | Children. | 299 | Logistic regression. | OR (drinkers vs non-drinkers) = 1.03 (0.63–1.69). OR (schistosome infection vs no infection) = 0.75 (0.32–1.74). |
| Friis (2009) [27]; cross-sectional | Regular drinkers, non-regular drinkers (regular drinking not defined). | Presence of eggs in stool sample. | Haemoglobin (g/L), serum ferritin (µg/L), and anaemia (Hb <120 g/L women, <130 g/L men). | Tuberculosis patients (15 – 85 years). | None reported. | 655 | Linear regression. | Schistosome infection was associated with decreased serum ferritin (p=0.001), and alcohol use was associated with increased serum ferritin (p=0.02). Alcohol use was also associated with higher haemoglobin (p=0.046). |
| Onile (2016) [28]; cross-sectional | Drinkers, non-drinkers (yes/no to alcohol consumption). | At least one egg per 10 mL urine sample (light infection = ≤50 eggs/10 mL, heavy infection = >50 eggs/10 mL). | Bladder pathology (defined as any of: abnormal bladder shape or wall thickness, bladder wall irregularities, bladder masses, bladder calcification, or polyps). | Adults (30–90 years). | Children. | 257 | Pearson's chi-squared. | Intensity of schistosome infection was associated with different types of bladder pathology (p=0.001). Alcohol consumption was not associated with bladder pathology (p=0.17). |

*(Continued)*

| Study and design | Alcohol use definition | Schistosome infection definition | Outcome | Population (age range) | Exclusion criteria | Number of participants | Statistical method | Result |
|---|---|---|---|---|---|---|---|---|
| Malenganisho (2007) [29]; cross-sectional | Categorised as never, <1 day/week (consumes alcohol less than weekly) and 1–7 day/week (consumes alcohol 1–7 days per week). | Presence of eggs in stool sample. | Haemoglobin (g/L), serum ferritin (µg/L), and anaemia (Hb <120 g/L women, <130 g/L men). | Participants were aged 14 and above (14 – 87 years). | Pregnant people. | 1498 | Linear regression. | Alcohol use was associated with higher Hb (p=0.01 for less than weekly; p=0.001 for weekly or more). Alcohol use was associated with higher serum ferritin (p=0.001 for less than weekly; p<0.001 for weekly or more). Schistosome infection was not significantly associated with either. |
| Li (2000) [30]; cohort | Moderate alcohol drinkers vs non-drinkers (moderate was defined as consuming alcohol weekly or more). | Presence of eggs in stool sample. | Improvement of PPF (lower grade according to the Cairo protocol). | Individuals in an S. japonicum endemic region of China. | None reported. | 193 | Logistic regression. | Alcohol was associated with lower odds of PPF improvement (OR = 0.29, 95% CI 0.08–0.91). Cure of heavy schistosome infection (defined as <100 epg at the end point) was associated with lower PPF (OR = 0.17, 95% CI 0.03–0.83). |
| Ssetaala (2015) [31]; case-control | Categorised as never, rarely (drank once per month or less), and regularly (at least once per week). | CAA value ≥32 pg/ml = positive. | HIV acquisition. | People in a high-HIV-risk fishing community (13 – 49 years). | None reported. | 200 | Logistic regression. | OR (schistosome infection vs no infection) = 1.23 (0.3–5.7) OR (regular alcohol use vs none) = 6.85 (1.3–37.4) OR (rare alcohol use vs none) = 9.21 (1.4–61.8). |
| Silva (2015) [32]; cross-sectional | Drinkers, non-drinkers (yes/ no to alcohol consumption). | Presence of eggs in stool sample. | Severe PPF (liver image pattern E or F). | Adults (18+) infected with S. mansoni. | History of cirrhosis, fatty liver disease, hepatitis B/C infection, infected with a species other than S. mansoni. | 79 | Logistic regression. | OR (alcohol use vs no alcohol use) = 0.2 (0.03–1.29). |
| McDonald (2018) [33]; randomised trial | Categorised as never, any alcohol consumption (1 or more drinks per week) and routine alcohol consumption (3 or more drinks per week). | Presence of eggs in stool sample. | Serum endo-toxin levels. | Pregnant women infected with S. japonicum. | None reported. | 353 | Linear regression. | Routine alcohol use was associated with increased serum endotoxin in cord blood (p=0.02). |
| Marinho (2020) [34]; cross-sectional | Drinkers, non-drinkers (yes/ no to alcohol consumption). | Presence of eggs in stool sample. | PPF (liver image pattern C–F). | Indigenous people living in Brazil (5 – 77 years). | None reported. | 148 | Logistic regression. | Schistosome infection was not associated (OR not reported). Alcohol use was associated with greater odds (OR = 3.54; 95% CI 1.4–8.8) of PPF. |

*(Continued)*

| Study and design | Alcohol use definition | Schistosome infection definition | Outcome | Population (age range) | Exclusion criteria | Number of participants | Statistical method | Result |
|---|---|---|---|---|---|---|---|---|
| Baya (2024) [35]; cohort | Drinkers, non-drinkers (yes/no to alcohol consumption). | Presence of eggs in stool sample. | S. mansoni–tuberculosis coinfection and treatment outcomes. | Residents of Mali (5 or more years) with confirmed pulmonary tuberculosis (14 – 76 years). | Culture-negative for Mycobacterium tuberculosis, or positive for multi-drug resistant tuberculosis. | 174 | Logistic regression. | Alcohol use was not associated with coinfection (OR = 2.02; 95% CI 0.55–7.50). Co-infection was associated with higher mortality rates (OR = 3.43; 95% CI 1.12–10.58) compared to tuberculosis mono-infected patients. |
| Houston (1993) [36]; cross-sectional | Reports collecting lifetime alcohol consumption, but does not define how this was measured. | Presence of eggs in stool sample. | PPF (Homeida classification 2–4). | Residents of the district who had spent their childhood there (15 – 88 years). | Non-permanent residents of the region. | 492 | Spearman correlation. | Neither schistosome infection nor alcohol use were correlated with PPF grade (p values not reported). |
| Andrade (2014) [37]; cross-sectional | No alcoholism vs alcoholism, defined as alcohol consumption above 20 g and 30 g daily for men and women, respectively. | Presence of eggs in stool sample. | Cirrhosis. | Adults (18+) with confirmed chronic hepatitis B infection. | Any other liver disease, receiving treatment for hepatitis B. | 406 | Logistic regression (an OR for current schistosome infection was calculated by constructing a 2×2 table). | OR (alcoholics vs non-alcoholics) = 2.46 (1.16–5.19). OR (schistosome infection vs no infection) = 2.28 (1.46–3.55). |
| Cota (2006) [38]; cohort | No alcohol abuse vs alcohol abuse, defined as >60 g ethanol per day. | Presence of eggs in stool sample. | PPF (defined as moderate or severe wall thickening observed in ultrasound). | Residents in an S. mansoni endemic area. | None reported. | 84 | Logistic regression. | OR (those who abuse alcohol vs those who do not) = 2.5 (1.6–4.1). |
| Barros Jr (1992) [39]; cross-sectional | Alcoholism versus no alcoholism (does not define alcoholism). | Presence of eggs in either stool sample or rectal biopsy. | Iron storage (scaled 1–4, where 4 is iron overload), and copper storage (considered both free and protein-bound copper). | Patients attending a clinic (8–90 years). | Presence of haemolytic anaemia, portacaval anastomosis, or a history of taking drinks or medicines containing iron or copper, family history of hemochromatosis. | 99 | Pearson's chi-squared, Fisher's exact test. | No association between schistosome infection or alcohol use and either of the outcomes (p values not reported). |

(Continued)

**Table 2.** (Continued)

| Study and design | Alcohol use definition | Schistosome infection definition | Outcome | Population (age range) | Exclusion criteria | Number of participants | Statistical method | Result |
|---|---|---|---|---|---|---|---|---|
| Wang (2025) [40]; cohort | Drinkers, non-drinkers (yes/no to alcohol consumption). | Presence of eggs in appendix tissue following biopsy. | Colorectal cancer. | Patients with appendicitis. | Obstruction of tumours. | 272 | Logistic regression, Pearson's chi-squared. | OR (schistosome infection vs no infection) = 5.09 (1.43–18.13). Does not report an OR for alcohol use, but reports that alcohol use was more common among those with schistosome infection (p=0.048). |
| Souza (2000) [41]; cross-sectional | Abstemious defined as individuals without a history of habitual alcohol intake; non-abstemious defined as those with a history of habitual alcohol intake (habitual not defined). | Presence of eggs in stool sample. | Portal hypertension. | Adults (19 – 70 years). | Chronic diseases other than schistosomiasis, hepatitis infection, using hepatotoxic drugs. | 54 | ROC analysis, Pearson's chi-squared. | Alcohol use did not impact the diagnostic accuracy of the biomarkers for portal hypertension (86% when alcohol users were included versus 96% when they were not). |
| Anjorin (2024) [42]; cross-sectional | Current drinkers consumed any alcohol in the past 12 months; current non-drinkers had not. | At least one egg per gram of stool. | PPF (liver image pattern C–F). | Individuals across 3 districts in rural Uganda (5 – 90 years). | Not resident for at least the previous 6 months of the year, drunk at time of survey, hospitalised. | 2834 | Logistic regression. | OR (current alcohol use vs not) = 1.09 (0.70–1.68) OR (schistosome infection vs no infection) = 0.893 (0.709–1.122). |
| Mohamed (1989) [43]; cohort | Drinkers, non-drinkers (yes/no to alcohol consumption). | Presence of eggs in stool sample. | Re-bleeding varices, mortality. | Patients who underwent sclerotherapy for oesophageal variceal bleeding (17 – 69 years). | None reported. | 100 | ORs calculated by constructing 2×2 tables from descriptive statistics. | For re-bleeding: infection vs no infection OR = 0.22 (0.07–0.65); alcohol use vs no alcohol use OR = 3.08 (0.18–52.6). For mortality: infection vs no infection OR = 0.07 (0.02–0.32). All alcohol users died. |
| Silva (2014) [44]; cross-sectional | Compares alcoholism vs no alcoholism, but does not define alcoholism. | Presence of eggs in stool sample. | Severe PPF (liver image pattern E or F). | Patients infected with S. mansoni. | Hepatitis infection, steatosis, ALD who received treatment in the last 3 months. | 203 | Logistic regression. | OR (comparing alcoholics to non-alcoholics) = 0.98 (0.34–2.92; p=0.831). |

studies did note that imaging was performed in hospitals or health clinics by an experienced operator. The WHO Niamey protocol was the most commonly used ultrasound grading method (66.7%, 6/9). Among these studies, four classified liver image patterns C to F as indicative of PPF. The two other PPF studies used a severity grouping that combined diffuse general fibrosis with PPF, classifying liver image patterns A-D as no or mild fibrosis, and liver image patterns E or F as indicating severe fibrosis. Two studies were published before the Niamey protocol - one study used the Cairo protocol, and the other study applied the Homeida classification. One study, published in 2006, did not specify a grading system but defined PPF as moderate to severe wall thickening of the portal branches of the liver visible on ultrasound. Portal hypertension was identified through endoscopy based on the presence of the clinical complication of gastro-oesophageal varices. In studies assessing anaemia as the main outcome, anaemia was diagnosed using complete blood count testing on venous blood samples, with thresholds defined according to WHO guidelines as haemoglobin <120 g/L for women and <130 g/L for men (studies did not include children). Serum ferritin, treated as a continuous variable and measured in μg/L, was also measured through blood tests. Cirrhosis was diagnosed by liver biopsy and scored using the METAVIR system. In the study examining colorectal cancer, the diagnostic modality was not specified.

## Associations of alcohol use and schistosome infection to disease outcomes

The associations between schistosome infection, alcohol use, and clinical outcomes were investigated across multiple disease domains, with considerable heterogeneity in how both exposures were defined. Statistical methods used and the associations observed are summarised in Table 2.

Hepatic outcomes were most frequently studied, with PPF being the most common outcome (9/21, 42.9%). The evidence was inconsistent and came from predominantly cross-sectional studies. Effect sizes ranged from negligible to strong for both exposures, with no consistent pattern. For example, one study reported higher odds of PPF with alcohol use but not schistosome infection [34], while another study found PPF severity to be associated with schistosome infection intensity but not alcohol use [24]. Other studies reported no association for either exposure [23,36,42,44]. A cohort study which evaluated treatment found that moderate alcohol consumption was associated with lower odds of PPF improvement following praziquantel, suggesting a potential exacerbating effect [30]. For the outcome of cirrhosis, both alcoholism and presence of PPF were found to be strong independent predictors [37].

Three studies (14.3%) investigated iron deficiency through markers including haemoglobin and serum ferritin. In these studies, alcohol use was consistently associated with higher haemoglobin and serum ferritin levels [27,29], whereas schistosome infection was linked to reduced ferritin in only one study [27], and showed no association in another [29].

Two studies (9.5%) examined infectious comorbidities. One reported that HIV acquisition was significantly associated with alcohol use but not schistosome infection [31]. A cohort study on tuberculosis found no association between alcohol use and *S. mansoni*-tuberculosis coinfection, though coinfection itself predicted higher mortality [35].

For the remaining studies, a distinct outcome was examined. These included bladder pathology (associated with schistosome infection intensity but not alcohol [28]), impaired renal function (associated with unhealthy drinking but not infection [25]), and colorectal cancer (associated with infection [40]). One study also found no significant effect of alcohol consumption on diagnostic accuracy of biomarkers for portal hypertension [41].

## Temporal relationships between exposures and diseases

Temporality, both in terms of prospective associations between the exposures and outcomes, and in sequential relationships between schistosomiasis and alcohol use, was rarely studied, as most studies (66.7%, 14/21) used cross-sectional designs (Table 1). Even among the studies which were not cross-sectional in design, the frequency of alcohol use assessments was either unclear or limited to a single baseline assessment. This approach rendered the analysis of alcohol use again as cross-sectional.

Five (23.8%) studies reported information which could be used to understand temporality either for individual exposures or for the sequential order of the two exposures. Andrade et al. modelled cirrhosis as the outcome to understand the role of schistosome infection and its related pathology [37]. Their multivariate regression revealed that the presence of PPF was the strongest predictor of cirrhosis (odds ratio [OR] 4.56, 95% CI 2.10 – 9.91), and that alcoholism was also a significant predictor (OR 2.46, 95% CI 1.16 – 5.19). A separate calculation using descriptive data showed that current schistosome infection was a weaker predictor of cirrhosis (OR 2.28, 95% CI 1.46 – 3.55). This suggests that the long-term damage associated with chronic infection is a more important indicator for cirrhosis than current infection status. A study by Marinho et al. considered PPF as the outcome, and reported an OR of 3.54 (95% CI 1.4 – 8.8) comparing drinkers to non-drinkers [34]. When stratified by age, the OR among individuals aged 30 years and older increased to 9.28 (95% CI 2.60 – 33.06) compared to non-drinkers aged 19 and younger.

Concerning sequential effects of schistosomiasis and alcohol use, Anjorin et al. proposed a lagged effect in the correlation between alcohol use and PPF, hypothesising that alcohol consumption during adolescence may increase the likelihood of developing PPF in adulthood, whereas schistosome exposure occurs before alcohol use in early childhood [42]. However, this study did not present data to assess this association quantitatively. Further supporting the sequence of schistosome exposure first then alcohol use later in life, Li et al. demonstrated that moderate to heavy alcohol intake exacerbates PPF following schistosome infection [30]. This conclusion was drawn from observations that participants who reported moderate alcohol consumption were less likely to show an improvement in PPF following treatment than those who did not (OR 0.29, 95% CI 0.08 - 0.91). Additionally, apart from PPF, Wang et al. observed higher alcohol consumption among patients with schistosome-associated appendicitis, implying either a sequential relationship where infection precedes alcohol-related pathology, or confounding [40].

## Statistical methods

Several statistical methods were used across the included studies, with some studies using more than one method. Seven studies (33.3%) presented only adjusted analyses, nine (42.9%) reported only unadjusted analyses, four (19.0%) presented both, and two (9.5%) studies provided sufficient descriptive data for odds ratios to be calculated using a 2 × 2 table. Regression analyses were used in 15 (71.4%) of the studies – 11 (73.3%, 11/15) employed logistic regression, three (20%, 3/15) used linear regression, and one used both. Other unadjusted methods included chi-squared tests, used in four studies, analysis of variance (ANOVA, one study) and receiver operating characteristic (ROC) curve analysis (one study). No studies included interaction terms. Two (9.5%) studies presented stratified analyses. Malenganisho et al. stratified by age and sex, and found that PPF was more prevalent among males compared to females within the same age groups, with this disparity becoming more pronounced with increasing age [24]. Marinho et al. stratified their analysis by age and alcohol consumption, finding that drinkers aged 20–29 had 2.5 times the odds of PPF compared to non-drinkers under 19 (95% CI 0.48 – 12.88), while drinkers aged 30 and older had 9.28 times the odds of PPF compared to the same reference group (95% CI 2.60 – 33.06) [34].

The most frequently included covariates in adjusted regression models were age (10/11), sex (7/10 as one study was female only), smoking (4/11), HIV status (3/11), and occupation (3/11). Table 3 presents the associations between common confounders and the two most common study outcomes – PPF and iron deficiency – stratified by schistosome infection diagnostic method and adjustment status. This stratification highlights that the evidence is mostly from studies using Kato-Katz microscopy, limiting comparisons across diagnostic techniques.

Age and sex (male versus female) were consistent covariates, with higher age and male sex frequently associated with increased odds of PPF. The influence of smoking and HIV status is less clear as they were reported in fewer studies. The variability in the reported associations shows how the unadjusted estimates for key exposures may be susceptible to confounding. Only four studies in this synthesis presented adjusted and unadjusted analyses for various outcomes, allowing for the assessment of confounding. Anjorin et al. observed that alcohol use was associated with increased odds of PPF

**Table 3. Direction of associations between common confounders and study outcomes. Associations are stratified by schistosome infection diagnostic modality and by whether the analyses were adjusted or unadjusted.**

| | Periportal fibrosis (9 studies) | Iron deficiency (3 studies) |
|---|---|---|
| **Schistosome infection diagnosed by Kato-Katz microscopy** | 8 studies | 3 studies |
| Age | | |
| Adjusted | + (3 studies)<br>no effect (1 study) | no effect (1 study) |
| Unadjusted | + (3 studies)<br>- (1 study)<br>no effect (1 study) | |
| Sex | | |
| Adjusted | + (1 study)<br>no effect (1 study) | + (1 study) |
| Unadjusted | + (4 studies)<br>no effect (1 study) | + (1 study) |
| Smoking | | |
| Adjusted | no effect (1 study) | - (1 study) |
| Unadjusted | + (1 study) | |
| HIV status | | |
| Adjusted | + (1 study) | + (1 study)<br>- (1 study) |
| Unadjusted | + (1 study) | |
| **Schistosome infection diagnosed by CCA urine test** | 1 study | 0 studies |
| Age | | |
| Adjusted | | |
| Unadjusted | + (1 study) | |
| Sex | | |
| Adjusted | | |
| Unadjusted | + (1 study) | |

Totals may exceed the number of studies per outcome because some studies reported both adjusted and unadjusted analyses.

Sex comparisons are male versus female.

A positive (+) symbol indicates a statistically significant positive association, where the risk factor is associated with an increased risk of the outcome.

A negative (−) symbol indicates a statistically significant negative association, whereby the risk factor is associated with a decreased risk of the outcome.

"No effect" indicates no statistically significant association.

in the unadjusted analysis (OR 3.63, 95% CI 2.66 – 4.91), but this association was not present in the adjusted model (OR 1.09, 95% CI 0.70 – 1.68), indicating confounding by factors like age and sex [42]. In contrast, Ssetaala et al. found that alcohol use remained associated with higher odds of HIV acquisition even after adjustment [31]. The OR for regular alcohol use compared to never drinkers increased from 3.49 (95% CI 1.6 – 7.8) in the unadjusted model to 6.85 (95% CI 1.3 – 37.4) in the adjusted model, and the OR for rare alcohol use rose from 1.92 (95% CI 0.8 – 4.8) to 9.21 (95% CI 1.4 – 61.8) - perhaps all signifying positive confounding. Baya et al. reported that none of age, gender, smoking, or alcohol use were associated with schistosome-tuberculosis coinfection in either the unadjusted or adjusted analyses [35]. Finally, Silva et al.

found no significant associations between alcohol use or age and the odds of severe PPF in either adjusted or unadjusted models [32]. Although sex was not included in their adjusted analysis, they reported an unadjusted OR comparing males to females of 2.85 (95% CI 1.0 – 8.28).

## Exposure interactions and biological mechanisms

Potential mechanisms linking alcohol use and schistosome infection to the clinical outcomes of interest were infrequently discussed by study authors or were not clearly specified in the model assumptions and motivation. For the few studies which discussed mechanisms, alcohol use and infection were presented individually rather than as co-occurring and potentially interacting exposures. Andrade et al. suggested that both schistosome infection and alcoholism contribute independently to liver cirrhosis progression, which may include synergistic effects of concurrent injuries, though interaction effects were not tested [37]. Similarly, Li et al. directly interpreted their findings as evidence that alcohol consumption "exacerbates schistosome-induced parenchymal morbidity", also suggesting a potential synergistic relationship, though again not formally tested [30]. Malenganisho et al. reported a higher proportion of PPF among alcohol drinkers compared to non-drinkers [24]. Although the difference was not statistically significant, they posited that alcohol may be a contributing factor to schistosome-induced liver morbidities and should be studied further. They also reported qualitatively that excessive alcohol consumption was predominantly observed in men, who also exhibited higher rates of PPF, suggesting a gender-dependent relationship.

Biological mechanisms were typically discussed for the exposures separately. Friis et al. discussed how alcohol consumption may enhance iron absorption due to the iron content of alcoholic drinks, or through increased iron solubilisation [27]. In contrast, schistosome infection was linked to intestinal bleeding, resulting in iron loss and reduced serum ferritin. Similarly, Malenganisho et al. found that alcohol use was positively associated with serum ferritin, likely reflecting increased iron absorption, while schistosome infection was linked to lower ferritin levels due to internal bleeding caused by infection [29].

## Bias and confounding

Reporting of bias and confounding varied across the studies. The most frequent source of bias reported by study authors was selection bias (28.6%, 6/21), which arose from either non-random sampling methods or non-random loss to follow-up. Several studies that did not explicitly mention selection bias also used sampling methods that could limit generalisability. For example, multiple studies recruited participants from hospitals or clinical settings, potentially causing people who were already unwell and seeking healthcare to be over-represented. In contrast, one study excluded people classified as drunkards. Survivor bias was reported in two studies. Misclassification bias was discussed in one study, where variability in egg output throughout the day was noted as a potential cause of misclassifying light infections as negative. Measurement error may have been present in studies which did not explicitly define their outcome measures, as variability in the definitions used by study staff could lead to inconsistencies and potential misclassification. However, it is also possible that standardised definitions were used and not reported. Recall bias was explicitly acknowledged in another two of the studies; however, all studies relied on self-reported alcohol consumption data, collected through interviews, questionnaires, or in a few cases, described simply as self-reported. The use of self-reported data introduces a potential for recall bias across all the studies, whether or not it was acknowledged by the study authors. The extent of social desirability bias may vary geographically, as the stigma surrounding alcohol use varies by region. In study settings where alcohol use is more stigmatised, under-reporting may be more likely, which could lead to smaller effect sizes being observed. This underreporting may be evident in one study of individuals aged five years and older in rural Uganda where the household head and spouse were asked to report alcohol use on behalf of all household members.

Study designs varied in representativeness of relevant populations for shared disease outcomes. The included studies covered a wide age range, although some explicitly excluded children. Several studies also lacked clear

age reporting, limiting the assessment of age representativeness. Study sizes were generally small, with only five (23.8%) including more than 500 participants. Some schistosomiasis-endemic countries such as Brazil, Tanzania, and Uganda were well-represented, however, others were underrepresented, such as China which only had two studies.

Additional confounders were rarely commented on by study authors, potentially due to the majority of analyses already being adjusted for key variables. The most commonly identified confounders by study authors were socioeconomic status (9.5%, 2/21) and infections (19.0%, 4/21), particularly HIV and hepatitis, or referred to more broadly as just "infections". An assessment of confounding was possible in the four studies mentioned previously that presented both unadjusted and adjusted analyses. The results from these studies demonstrate that, for the studies presenting only unadjusted analyses, confounding by factors such as age, sex, and socioeconomic status may be impacting the observed effect sizes. Other potential confounders, including occupation, water contact behaviour, and smoking status, were identified through the data extraction where confounding was not reported on by study authors, depending on the population and clinical outcome being studied.

## Discussion

The growing double burden of infectious and non-communicable diseases presents an important public health challenge, particularly in LMICs [45]. Addressing this challenge requires an understanding of the mechanistic relationships between concurrent risk factors and shared disease presentations. This review synthesises the findings from 21 studies of 11,794 individuals across nine countries to investigate the clinical outcomes associated with co-occurring schistosome infection and alcohol use. Our review found that studies tend to focus on schistosome-specific pathologies where alcohol may be a modifying factor and highlights the need for future research to incorporate temporal analyses to better understand the longitudinal effects of these exposures. Although the initial hypothesis of this review was that a synergistic relationship exists between alcohol use and schistosome infection, our synthesis suggests that evidence exists for both synergistic and antagonistic effects.

### Disease outcomes and hypothesised pathways

The relationship between alcohol use and schistosome infection has been investigated primarily within the context of hepatic morbidity, namely PPF, since both exposures are well-established aetiological agents of liver disease. Schistosome infection was reported to be the main driver of PPF, with alcohol use presented as a significant effect modifier which could exacerbate pathology in certain contexts [34,37]. However, there is a need for future studies to more clearly define chronic or heavy alcohol use. The dominance of schistosome-specific pathologies in the literature means that outcomes associated with alcohol use may have been missed. For instance, hepatocellular carcinoma is considered one of the final manifestations of ALD, and infection with *S. japonicum* can also independently cause hepatocellular carcinoma [46]. Similarly, steatosis (fatty liver) is a common and reversible early-stage manifestation of ALD, but it is not causally attributable to schistosome infection alone [47]. It therefore remains unclear whether schistosome infection modifies the risk or progression of alcohol-specific outcomes.

A synergistic interaction between schistosome infection and alcohol consumption, leading to accelerated liver fibrosis, is mechanistically plausible. Egg deposition during schistosome infection initially causes focal injury through granuloma formation and inflammation in the portal veins, and then alcohol metabolism induces diffuse hepatocellular damage and oxidative stress [48–51]. In co-exposed individuals, the toxicity of alcohol metabolism in hepatocytes near and around the portal vein could exacerbate schistosome-induced damage and cause more severe manifestations of PPF. Findings from murine models also provide evidence that supports this synergistic pathway. Experimental co-exposure in mice has resulted in more severe liver pathology, including larger granulomas and increased fibrosis, than would be expected from each exposure alone [16,17].

There is also experimental evidence to support an antagonistic pathway between schistosome infection and alcohol use through immune modulation via alcohol-induced immune suppression. Chronic alcohol intake has been reported to have significantly reduced the size of schistosome-induced granulomas in mice [52]. The authors of this study suggest that the impairment of cell-mediated immunity due to alcohol use dampens the host inflammatory response to schistosome eggs. An antagonistic pathway is also possible in the development of steatosis, although this was not an outcome explored in any of the included studies. There is preliminary mechanistic evidence from mice that schistosomes rely on host high-density lipoproteins for egg development and continuation of the parasite life cycle [53]. Experimental data from mice also found reduced steatosis in schistosome-infected mice, even following exposure to alcohol [16]. In human studies, schistosome infection has been associated with dyslipidaemia, reflecting disturbances in host lipid metabolism and transport [54,55]. Therefore, an antagonistic interaction between alcohol use and schistosome infection mediated through regulation of hepatic lipid metabolism is plausible. Another antagonistic relationship between schistosome infection and alcohol use may be possible for iron-related outcomes. A potential pathway for this is through regulation of hepcidin – a hormone that is primarily produced in the liver and plays a key role in the regulation of blood iron levels [56]. Alcohol consumption typically suppresses hepcidin synthesis, which increases iron absorption into the blood, whereas the immune response to schistosome infection has been shown to increase hepcidin synthesis, resulting in iron absorption becoming more tightly regulated [57,58]. This pathway may explain the differences in ferritin patterns observed in the included studies [27,29].

Although both synergistic and antagonistic relationships have support from murine studies, their translational relevance to human disease is limited. Firstly, experimental models typically administer the exposures in close temporal proximity in adult animals [16,17,52], which is not an accurate reflection of exposure in humans – schistosome infection typically begins in early childhood, before alcohol use is initiated later in life. Furthermore, the controlled dose of each exposure in mice does not reflect the heterogeneity in timing and dose that occurs in human populations, for example when considering behaviours such as binge drinking. Therefore, murine models alone would not be able to resolve which kind of interaction prevails for humans. Instead, prospective human studies over much longer periods of time, such as those underway in SchistoTrack [59], are needed.

## Temporality

The current literature is majorly limited in its ability to establish temporal relationships between schistosome infection, alcohol use, and clinical outcomes. The predominance of cross-sectional studies, alongside the common assessment of alcohol use at a single timepoint, assumes that exposures and outcomes are both static and concurrent. This approach fails to capture possible changes in exposure intensity, duration, and sequence that can occur over the life-course and alter their interactions. Additionally, the reliance on binary measures of current schistosome infection limits the ability to assess the impact of chronic and cumulative infection on outcomes such as cirrhosis. However, the review did identify some studies which can be used to comment on temporality and identify future directions. The stratified analysis by Marinho et al., which found greatly higher odds of PPF among older drinkers, could reflect increased vulnerability to liver damage as enzyme activity decreases with age [60]. Another possible explanation is that drinking patterns change over the life of an individual – older people might drink more than younger people in the study population. This finding aligns with the observations reported by Li et al., which suggests that alcohol consumption can alter the trajectory of disease progression and response to treatment [30]. Together, these observations highlight the necessity for longitudinal studies which are designed to elucidate the complexity of changing exposure dynamics over time.

A thorough understanding of temporality requires a consideration of the life-course patterns of both schistosome infection and alcohol use. Schistosome infection prevalence and intensity peaks among young children, however the severe pathology associated with chronic infections, such as PPF, tends to manifest in adulthood [61]. Previous work by Ewuzie

et al. reported that current schistosome infection is only weakly linked to PPF, and may not be the best proxy indicator of cumulative infection burden or morbidity risk [12]. Similarly, it has been found that current water contact, the key exposure for infection, is poorly correlated with infection status [62], which reflects the complex immune response to infection and the resistance to reinfection that can build over time. These findings highlight how morbidity results from the cumulative, chronic nature of infection rather than from a single point of current exposure.

Alcohol consumption also follows its own temporal trajectory. While initiation of alcohol use tends to occur during adolescence [5,63], patterns of hazardous drinking and alcohol use disorders frequently peak in middle age [64,65], years or even decades after initial schistosome infection was established. This staggered exposure timing may suggest that alcohol-induced liver injury would develop years after schistosome infection, and alcohol use may compound already established schistosome-induced liver damage. Given the timing of schistosome exposure and alcohol use in later life, future studies should consider monitoring populations starting relatively young and first tracking schistosome infection history, then the development of alcohol use behaviours to understand shared disease outcomes.

## Limitations

There are some limitations to our study. As with all scoping reviews, the findings are constrained by the quality and completeness of reporting in the included studies. In particular, reporting of alcohol use varied markedly in quality, although alcohol use was often not the primary focus of the study, so this may be an unfair expectation. Future studies should employ standardised, clinically meaningful definitions of alcohol use. Previous research has suggested that categorisations of alcohol use should be based on biological consequence in relation to the disease outcome – for example, combining lifetime abstainers with very low-level alcohol users, rather than a broad drinker versus non-drinker divide, which risks diluting the effect estimates for higher alcohol use groups [10]. A further limitation relates to the interpretation of joint effects between alcohol use and schistosome infection. None of the studies reported formally testing interaction terms – instead, alcohol use and schistosome infection were measured independently using exposure classifications that are subject to measurement error. In the context of joint exposure assessment, correlated misclassification, arising from underreporting of alcohol consumption or false-negative schistosome infection classification when single-sample microscopy is used, may have resulted in individuals being misclassified into lower exposure categories for both factors. Consequently, some observed associations may reflect measurement error rather than true joint biological effects. Future studies could mitigate these limitations by using repeated measures of alcohol consumption in a subsample of participants, and by applying more reliable schistosomiasis exposure assessment strategies, such as repeated stool sampling or the use of more sensitive antigen-based diagnostics. Additionally, as the majority of the studies examined *S. mansoni*, there are not enough studies involving other schistosome species to allow for an assessment of species-specific effects on clinical outcomes. Selection bias was also evident, with several studies recruiting from clinical settings and one study excluding heavy drinkers, resulting in study populations that are not likely to be representative. As mentioned previously, the analytical approaches used in the included studies also limit the review. Only four studies reported both adjusted and unadjusted analyses which allowed for the assessment of confounding. This suggests that confounding will impact the associations observed in other studies, especially those reporting only unadjusted analyses.

## Conclusion and policy implications

This review aimed to examine alcohol use and schistosome infection as co-occurring exposures for clinical outcomes, particularly those related to liver disease. The overlap of these exposures in endemic settings represents an important and understudied area of multi-morbidity that requires more attention in both research and public health policy. Future research should prioritise longitudinal designs that allow for the study of temporal and causal relationships between alcohol use, schistosome infection, and clinical outcomes. Standardised and biologically meaningful definitions of alcohol

use, in addition to improved diagnostic methods for schistosome infection, will improve the quality of data and allow more generalisable conclusions. Alcohol use was consistently reported as an effect modifier for schistosome-related pathology, and so management of schistosomiasis in endemic regions should incorporate some level of alcohol use assessment to identify those at risk of developing more severe disease. With additional studies, integrated public health approaches to control schistosome infection and alcohol use may improve liver outcomes for low-middle income countries.

## Supporting information

**S1 File. PRISMA-ScR checklist.**
(DOCX)

**S2 File. Search strings used and number of articles returned for each of the databases.**
(DOCX)

**S3 File. Completed data extraction form for all studies included in the review.**
(XLSX)

**S4 File. List of excluded full texts, with reasons for exclusion.**
(CSV)

## Acknowledgments

We thank Nia Roberts at the Bodleian Libraries for helping with the initial stages of the review. We thank Fabian Reitzug, Margarida Bica, and Zhaoyu Guo for translating full texts that were not in English. We also thank the Chami Group for helping with the retrieval of full text articles and for their feedback during group meetings where this work was presented.

## Author contributions

**Conceptualization:** Bethany Lyne, Sarah Lewington, Goylette F. Chami.

**Data curation:** Bethany Lyne, Max M Lang.

**Formal analysis:** Bethany Lyne.

**Funding acquisition:** Bethany Lyne, Goylette F. Chami.

**Methodology:** Bethany Lyne.

**Resources:** Goylette F. Chami.

**Supervision:** Bethany Lyne, Sarah Lewington, Goylette F. Chami.

**Writing – original draft:** Bethany Lyne.

**Writing – review & editing:** Bethany Lyne, Max M Lang, Sarah Lewington, Goylette F. Chami.

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
