## [Decision Letter · Decision Letter 0]

18 Dec 2025

PNTD-D-25-01763

Clinical outcomes associated with schistosome infection and alcohol use: a systematic scoping review

Dear Dr. Chami,

Thank you for submitting your manuscript to PLOS Neglected Tropical Diseases. After careful consideration, we feel that it has merit but does not fully meet PLOS Neglected Tropical Diseases's publication criteria as it currently stands. Therefore, we invite you to submit a revised version of the manuscript that addresses the points raised during the review process.

Please submit your revised manuscript within by Feb 16 2026 11:59PM. If you will need more time than this to complete your revisions, please reply to this message or contact the journal office at plosntds@plos.org. Please include the following items when submitting your revised manuscript:

We look forward to receiving your revised manuscript.

Kind regards,

Justin Komguep Nono, PhD

Academic Editor

Jong-Yil Chai

Section Editor

Shaden Kamhawi

co-Editor-in-Chief

Paul Brindley

co-Editor-in-Chief

**Additional Editor Comments :**

In general, the reviewers were positive with regards to your submission. The research question on the links between schistosomiasis and alcohol use as co-occurring risk factors for liver and other systemic pathologies is timely and the summary of current literature on the topic proposed here is clearly presented and adhere to scoping reviews’ standards.

However, the conceptual and analytical contributions remain modest and would greatly benefit from

i) deeper insights building on the descriptive mapping presented here,

ii) a more impactful reorganization of the text and tables as well as

iii) a better description of the included studies strengths and weaknesses for a clearer presentation of the limitations of the proposed systematic review to the reader.

In light of the above points, detailed in the received reviews on this submission (below this email), we would like to invite the resubmission of a significantly revised version that takes into account all reviewers' comments.

**Journal Requirements:**

1) Please upload all main figures as separate Figure files in .tif or .eps format. For more information about how to convert and format your figure files please see our guidelines:

2) Tables should not be uploaded as individual files. Please remove these files as they should only be included in the manuscript file as editable, cell-based objects. For more information about how to format tables, see our guidelines:

https://journals.plos.org/plosntds/s/tables

3) Please amend your detailed Financial Disclosure statement. This is published with the article. It must therefore be completed in full sentences and contain the exact wording you wish to be published.

4) Please ensure that the funders and grant numbers match between the Financial Disclosure field and the Funding Information tab in your submission form. Note that the funders must be provided in the same order in both places as well. Currently, the order of the funders is different in both places.

5) As required by our policy on Data Availability, please ensure your manuscript or supplementary information includes the following:

This information can be included in the main text, supplementary information, or relevant data repository. Please note that providing these underlying data is a requirement for publication in this journal.

**Comments to the Authors:**

**Please note that one review is uploaded as an attachment.**

**Reviewers' Comments:**

Reviewer's Responses to Questions

**Key Review Criteria Required for Acceptance?**

**Methods**

-Are the objectives of the study clearly articulated with a clear testable hypothesis stated?

-Is the study design appropriate to address the stated objectives?

-Is the population clearly described and appropriate for the hypothesis being tested?

-Is the sample size sufficient to ensure adequate power to address the hypothesis being tested?

-Were correct statistical analysis used to support conclusions?

-Are there concerns about ethical or regulatory requirements being met?

Reviewer #1: See attached file

Reviewer #2: The review by Lyne et al. summarizes the results of several studies to conclude that there is a synergy between alcohol consumption and schistosome infection in terms of liver pathology. However, most of the included studies did not explicitly measure or model joint exposure or effect modification.

Therefore, it is critical to note that systematic, correlated classifications of alcohol consumption and schistosome infection in primary studies undermine the reliability of joint exposure categories used to infer synergies for liver disease.

Without correction, the observed patterns may reflect measurement error rather than true joint effects. Self-reported underreporting of alcohol consumption and false-negative results in single Kato-Katz samples may together lead to individuals being classified into lower exposure categories for both factors.

The evidence summarized in the review is predominantly cross-sectional (Table 1), and alcohol consumption was typically measured at a single point in time using heterogeneous, self-reported definitions (Table 2). The discussion explicitly acknowledges the lack of temporal information and the limited ability to assess the sequence and duration of exposures, which is critical for determining whether the combined effects are synergistic over the lifespan.

Please comment on that.

Reviewer #3: 1. study objectives are clearly articulated. As a scoping review the authors are reviewing existing evidence assessing schistosomiasis and alcohol consumption on clinical outcomes

2. The hypothesis needs to be stated

3. 21 papers were selected for the review, authors need to include population numbers

4. The selected papers provide insight into any existing associations; however, the authors do appreciate limitations of this low number and justify the need for further studies.

5. Statistical analysis adequate

6. There were no ethical requirements for this review

**Results**

-Does the analysis presented match the analysis plan?

-Are the results clearly and completely presented?

-Are the figures (Tables, Images) of sufficient quality for clarity?

Reviewer #1: See attached file

Reviewer #2: The PRISMA flow in Figure 1 and the study-level details in Table 1 and Table 2 support a qualitative synthesis confirming synergistic hepatic effects, but the review does not perform a quantitative meta-analysis or meta-regression to formally estimate and test additive or multiplicative interactions between alcohol consumption and schistosome infection with regard to hepatic outcomes. While individual studies demonstrate strong interactions, there is no comprehensive meta-analysis summarizing these effect sizes. Without pooled effect sizes and interaction tests, the conclusion of synergistic effects is based on narrative patterns and isolated study results that are susceptible to bias and heterogeneity at the study level. A formal meta-analysis with random effects and a meta-regression would quantify the variance between studies and test whether the joint exposure effect exceeds the sum or product of the individual exposure effects.

Perhaps the authors can comment on this and supplement the proposed calculations.

Reviewer #3: 1. The analysis plan follows: the Preferred Reporting Items for Systematic reviews and Meta-Analyses extension for Scoping Reviews (PRISMA-ScR) Checklist.

2. Figure 1- the authors state that there was 1 additional study from other sources, can they specify the source

3. Figure 1: Under screening for 164 studies excluded: the breakdown of various studies does to add up to 164, if there is overlap this needs correcting

4. Table 2: include population numbers and age band where available for the 21 papers in the table

**Conclusions**

-Are the conclusions supported by the data presented?

-Are the limitations of analysis clearly described?

-Do the authors discuss how these data can be helpful to advance our understanding of the topic under study?

-Is public health relevance addressed?

Reviewer #1: See attached file

Reviewer #2: yes

Reviewer #3: 1. the discussion sufficiently addresses disease outcomes, however in Line 180- the authors give examples of chronic, heavy alcohol use among the older age group. None of the studies clearly define chronic or heavy alcohol use, nor do they specify how these categories were established (Anjorin et al., 2024). It would be beneficial if the authors could provide clarification on these definitions.

2. The methods used to diagnosis vary between majority of papers using microscopy, two antigen-based test and one biopsy how does this impact on study outcomes? any limitations?

3. Both limitations and Policy implications of this study have been addressed

**Editorial and Data Presentation Modifications?**

Reviewer #1: See attached file

Reviewer #2: (No Response)

Reviewer #3: (No Response)

**Summary and General Comments**

Reviewer #1: See attached file

Reviewer #2: (No Response)

Reviewer #3: The authors have undertaken a systematic scoping review to address the how schistosome infection and alcohol use influence health outcomes specifically: clinical outcomes, biological and temporal pathways.

The study is a relatively new area with paucity of data. The authors were able to identify 21 publications selected over the past 32 years. These papers represent key schistosomiasis endemic regions.

Overall, Lyne et al address a very important area of work that still remains highly understudied and this merits future work. Having said this, they have a preprint Lyne, B., Besong, M. E., Nabatte, B., Tinkitina, B., Oryema, J. B., Kabatereine, N. B., Lewington, S., & Chami, G. F. “Co-occurrence of alcohol use and Schistosoma mansoni infection: prevalence, patterns, and risk factors in rural Uganda.” medRxiv preprint (2025). Can this not be referenced as unpublished but can shed more light on general conclusions for this paper?

Other minor comments to consider:

1) In populations studied- were they long term exposed/chronic or recent infections and what impact did this have on infection vs cirrhosis?

2) Were there any differences in species studied on study outcomes?

3) Introduction section:

a) Line 77- limited data on mortality rate of schistosomiasis unknown. This is not wholly true some estimates put it approximately at 11,792 deaths per year globally (WHO, 2023). However, this is estimation is due to hidden pathologies like liver and kidney failure, bladder cancer, and ectopic pregnancies related to female genital schistosomiasis.

b) Line 82: Reference (7) pertains specifically to a study specifically on HIV and does not address the co-occurrence of schistosomiasis and alcohol use in the fishing communities examined. This should be corrected

c) Line 84: Reference (8) represents a study specifically assessing liver cirrhosis and again there is no mention of occupational stresses of fishing promoting alcohol use as a coping mechanism. Again, correct this.

PLOS authors have the option to publish the peer review history of their article (what does this mean?). If published, this will include your full peer review and any attached files.

Reviewer #1: No

Reviewer #2: **Yes:** Martina Sombetzki

Reviewer #3: **Yes:** Lucy Ochola

**Figure resubmission:**

While revising your submission, we strongly recommend that you use PLOS’s NAAS tool (https://ngplosjournals.pagemajik.ai/artanalysis) to test your figure files. NAAS can convert your figure files to the TIFF file type and meet basic requirements (such as print size, resolution), or provide you with a report on issues that do not meet our requirements and that NAAS cannot fix.  After uploading your figures to PLOS’s NAAS tool - https://ngplosjournals.pagemajik.ai/artanalysis, NAAS will process the files provided and display the results in the "Uploaded Files" section of the page as the processing is complete. If the uploaded figures meet our requirements (or NAAS is able to fix the files to meet our requirements), the figure will be marked as "fixed" above. If NAAS is unable to fix the files, a red "failed" label will appear above. When NAAS has confirmed that the figure files meet our requirements, please download the file via the download option, and include these NAAS processed figure files when submitting your revised manuscript.
---

## [Decision Letter · Decision Letter 1]

20 Apr 2026

PNTD-D-25-01763R1

Clinical outcomes associated with schistosome infection and alcohol use: a systematic scoping review

Dear Dr. Chami,

Thank you for submitting your manuscript to PLOS Neglected Tropical Diseases. After careful consideration, we feel that it has merit but does not fully meet PLOS Neglected Tropical Diseases's publication criteria as it currently stands. Therefore, we invite you to submit a revised version of the manuscript that addresses the points raised during the review process.

We look forward to receiving your revised manuscript.

Kind regards,

Robert Adamu SHEY, Ph.D.

Guest Editor

Jong-Yil Chai

Section Editor

Shaden Kamhawi

co-Editor-in-Chief

Paul Brindley

co-Editor-in-Chief

**Journal Requirements:**

1) As required by our policy on Data Availability, please ensure your manuscript or supplementary information includes the following:

**Reviewers' comments:**

Reviewer's Responses to Questions

**Key Review Criteria Required for Acceptance?**

**Methods**

-Are the objectives of the study clearly articulated with a clear testable hypothesis stated?

-Is the study design appropriate to address the stated objectives?

-Is the population clearly described and appropriate for the hypothesis being tested?

-Is the sample size sufficient to ensure adequate power to address the hypothesis being tested?

-Were correct statistical analysis used to support conclusions?

-Are there concerns about ethical or regulatory requirements being met?

Reviewer #3: (No Response)

Reviewer #4: -Are the objectives of the study clearly articulated with a clear testable hypothesis stated?

Yes

-Is the study design appropriate to address the stated objectives?

Yes

-Is the population clearly described and appropriate for the hypothesis being tested?

Yes

-Is the sample size sufficient to ensure adequate power to address the hypothesis being tested?

Not applicable

-Were correct statistical analysis used to support conclusions?

Not applicable

-Are there concerns about ethical or regulatory requirements being met?

No.

Reviewer #5: My critical comments are about wording in the abstract, some grammatical comments, and corrections to make to the references, all of which I’ll put in the sections about Methods and Results:

The Abstract (Methods) refers to this review as a “systematic review”. As it is not a “systematic review”, please correct that to “scoping review”. Please also add the word “frequently” to this sentence in the Results section: “The most frequently reported clinical outcomes were…”.

METHODS/ Database search and selection criteria (p.6). Please include information about the platforms from which you accessed the databases. For example, did you search the Web of Science (All Databases), or the Web of Science Core Collection? Did you search Global Health via Ovid?

Thank you for explaining the rationale behind the search strategy to the previous peer reviewer. Please clarify which fields in the databases you searched (in reference to the search strategies available in Supplementary File S2). For example, did you search All Fields for the PubMed search, as you did for the Ovid databases (.mp.), and did you search the TOPIC field in Web of Science? Why did you not truncate alcohol* to include papers about alcoholism or alcoholics? Do you think that affects the results? You risk missing papers, eg. 89 papers in PubMed for the search for ‘alcohol’ not ‘alcohol*’: https://pubmed.ncbi.nlm.nih.gov/?term=%28%28alcohol*+AND+schistosom*%29%29+NOT+%28alcohol+AND+Schistosom*%29&sort=date

**Results**

-Does the analysis presented match the analysis plan?

-Are the results clearly and completely presented?

-Are the figures (Tables, Images) of sufficient quality for clarity?

Reviewer #3: (No Response)

Reviewer #4: -Does the analysis presented match the analysis plan?

Not applicable

-Are the results clearly and completely presented?

Yes

-Are the figures (Tables, Images) of sufficient quality for clarity?

Yes

Reviewer #5: Table 2: please add the correct citation details for each reference in the table, eg. Kavishe et al, 2021 [25].

Please then be consistent in your use of the number citation style instead of using the Harvard citation style (p.19, the whole Associations of alcohol use and schistosome infection to disease outcomes section), and make sure all references are cited. For example, p.19 line 358 you do not cite the study by Andrade et al, but p.20 (line 366) you do cite the study by Marinho et al.

**Conclusions**

-Are the conclusions supported by the data presented?

-Are the limitations of analysis clearly described?

-Do the authors discuss how these data can be helpful to advance our understanding of the topic under study?

-Is public health relevance addressed?

Reviewer #3: (No Response)

Reviewer #4: -Are the conclusions supported by the data presented?

Yes

-Are the limitations of analysis clearly described?

May need improvement. See below.

-Do the authors discuss how these data can be helpful to advance our understanding of the topic under study?

Yes.

-Is public health relevance addressed?

Yes.

Reviewer #5: The argument is persuasive in the Conclusions, and public health relevance is addressed.

**Editorial and Data Presentation Modifications?**

Reviewer #3: (No Response)

Reviewer #4: (No Response)

Reviewer #5: References (pp.28-9): Please provide more citation details for [7] and [14].

Minor Revision.

**Summary and General Comments**

Reviewer #3: (No Response)

Reviewer #4: The original and the revision was read with the discussion between reviewers. All the issues pointed out by the first reviewers were addressed and the manuscript was improved overall.

I suggest to add a limitation of including only English articles which might have led to missing studies.

Reviewer #5: Thank you for providing me with the opportunity to review this paper. I am peer reviewing this paper in my professional capacity as an information specialist, adding to the reports that others have already provided you with.

There is a gap in the literature for an evidence synthesis study about schistosome infection and alcohol use, and this scoping review provides valuable insights about future directions for research. This is original research which is of importance to researchers and policy makers, as well as being of interest to researchers and practitioners outside the field.

PLOS authors have the option to publish the peer review history of their article (what does this mean?). If published, this will include your full peer review and any attached files.

**Do you want your identity to be public for this peer review?** For information about this choice, including consent withdrawal, please see our Privacy Policy.

Reviewer #3: No

Reviewer #4: **Yes:** kentaro iwata

Reviewer #5: No

**Figure resubmission:**

---

## [Editor Report · Decision Letter 2]

12 May 2026

Dear Dr Chami,

We are pleased to inform you that your manuscript 'Clinical outcomes associated with schistosome infection and alcohol use: a systematic scoping review' has been provisionally accepted for publication in PLOS Neglected Tropical Diseases.

Best regards,

Robert Adamu SHEY, Ph.D.

Guest Editor

Jong-Yil Chai

Section Editor

Shaden Kamhawi

co-Editor-in-Chief

Paul Brindley

co-Editor-in-Chief

---

## [Editor Report · Acceptance letter]

Dear Dr Chami,

We are delighted to inform you that your manuscript, "Clinical outcomes associated with schistosome infection and alcohol use: a systematic scoping review," has been formally accepted for publication in PLOS Neglected Tropical Diseases.

Best regards,

Shaden Kamhawi

co-Editor-in-Chief

Paul Brindley

co-Editor-in-Chief
